# Socioeconomic and behavioral determinants of cardiovascular diseases among older adults in Belgium and France: A longitudinal analysis from the SHARE study

Hamid Yimam Hassen[1]*, Hilde Bastiaens[1,2], Kathleen Van Royen[1,3], Steven Abrams[2,4]

1 Department of Primary and Interdisciplinary Care, Faculty of Medicine and Health Sciences, University of Antwerp, Antwerp, Belgium, 2 Global Health Institute, Faculty of Medicine and Health Sciences, University of Antwerp, Antwerp, Belgium, 3 Department of Communication Studies, Faculty of Social Sciences, University of Antwerp, Antwerp, Belgium, 4 Interuniversity Institute for Biostatistics and statistical Bioinformatics, Data Science Institute, Hasselt University, Diepenbeek, Belgium

* Hamid.Hassen@uantwerpen.be

**Data Availability Statement:** The data underlying the results presented in the study are available

## Abstract

Despite advances in the healthcare system, cardiovascular diseases (CVDs) are still an important public health problem with disparities in the burden within and between countries. Studies among the adult population documented that socioeconomic and environmental factors play a role in the incidence and progression of CVDs. However, evidence is scarce on the socioeconomic determinants and the interplay with behavioral risks among older adults. Therefore, we identified socioeconomic and behavioral determinants of CVDs among older adults. Our sample consisted of 14,322 people aged 50 years and above from Belgium and France who responded to the waves 4, 5, 6 and/or 7 of the Survey of Health Ageing and Retirement in Europe. The effect of determinants on the occurrence of CVD was examined using a Generalized Estimating Equation (GEE) approach for binary longitudinal data. The overall rate of heart attack was 8.3%, which is 7.6% in Belgium and 9.1% in France. Whereas, 2.6% and 2.3% in Belgium and France, respectively, had experienced stroke. In the multivariable GEE model, older age [AOR: 1.057, 95%CI: 1.055–1.060], living in large cities [AOR: 1.14, 95%CI: 1.07–1.18], and retirement [AOR: 1.21, 95%CI: 1.16–1.31] were associated with higher risk of CVD. Furthermore, higher level of education [AOR: 0.82, 95%CI: 0.79–0.90], upper wealth quantile [AOR: 0.82, 95%CI: 0.76–0.86] and having social support [AOR: 0.81, 95%CI: 0.77–0.84] significantly lowers the odds of having CVD. A higher hand grip strength was also significantly associated with lower risk of CVD [AOR: 0.987, 95%CI: 0.984–0.990]. This study demonstrated that older adults who do not have social support, live in big cities, belong to the lowest wealth quantile, and have a low level of education have a higher likelihood of CVD. Therefore, community-based interventions aimed at reducing cardiovascular risks need to give more emphasis to high-risk retired older adults with lower education, no social support and those who live in large cities.

from the SHARE Research Data Center (http://www.share-project.org/data-documentation/share-data-releases.html).

**Funding:** This study utilized data from the SHARE wave 4 to 7. The SHARE data collection is primarily funded by the European Commission (Horizon 2020), the US National Institute on Aging, and various national sources (For a full list of funding institutions, see http://www.share-project.org). This specific analysis is supported by EU funded SPICES project (Grant agreement ID: 733356). All funders had no role in the study design, data collection and analysis, decision to publish, or preparation of the manuscript.

**Competing interests:** The authors have declared that no competing interests exist.

# Introduction

Cardiovascular diseases (CVDs) are the leading cause of morbidity and mortality worldwide and constitute a major burden on the healthcare system in all countries [1]. In 2017, an estimated 422.7 million prevalent cases, 17.9 million deaths, and 366 million disability adjusted life years (DALYs) were attributed to CVDs [2]. In Europe, nearly half of all deaths are related to CVDs so being responsible for more deaths than any other condition [3, 4]. In Belgium and France, CVDs account for 28% and 24% of all deaths respectively [5].

Despite advances in the primary and secondary prevention, there are still disparities in the CVD burden within and across countries, some segment of the population being at a higher risk [6–8]. Previously, epidemiological studies have focused on identifying and modifying individual risk factors. As a result, several health conditions and behavioral factors such as tobacco use, unhealthy diet, alcohol consumption and physical inactivity were identified to be related with CVDs. The reduction of such unhealthy lifestyles and early detection and treatment of risk factors are effective to reduce the burden. Hence, improvements have been observed in the reduction of CVD morbidities and its associated premature mortality among adults. However, many cardiovascular risks still remain as a major public health issue at different rates globally. The burden of CVDs highly varied across segments of the population, older age, the least affluent and deprived communities are disproportionately affected [9–11]. Socioeconomic and environmental factors, including income, living condition, level of education also play an important role in the development, progression and outcomes of CVDs [12–15].

As part of the efforts to narrow disparities, the World Health Organization (WHO) established a Commission on Social Determinants of Health and developed a conceptual framework for action [16]. The framework emphasizes that social determinants as well as resources to prevent illness, are not distributed randomly throughout human society and such disparities need to be identified. Albeit some studies assessed the socioeconomic determinants of CVDs in the general population, the evidence specific to older adults is scant. Older adults have a different lifestyle and social dynamics compared to younger counterparts [17]. As age increases, the shrinkage of social networks through deaths of friends puts older people at risk of social isolation and loneliness, which might be associated with the occurrence of CVDs in those individuals [18].

Identifying the socioeconomic and behavioral factors of CVDs and evaluating the moderation effect in between could help in improving intervention strategies to reduce the burden of CVD in the older population. The EU Horizon 2020 funded project named SPICES–Scaling-up Packages of Interventions for cardiovascular diseases in Europe and Sub-Saharan Africa, aims to reduce CVD risks among vulnerable populations through risk profiling and coaching intermediate-risk participants. The current study supports the identification of target population and area of intervention among high-risk older adults in Belgium and France. Hence, in this study, we identified the socioeconomic determinants and behavioral risks of CVDs among older adults in Belgium and France, using a longitudinal data collected in the Survey of Health, Ageing and Retirement in Europe (SHARE) study. We also assessed the moderation and interplay of socioeconomic, behavioral and clinical risk factors on CVD, particularly heart attack and stroke.

# Methods and materials

## Study design and participants

We used the longitudinal dataset from the SHARE survey, which collects information on health status, behavioral risks, socioeconomic condition, and social networks of individuals

aged 50 years and above and their partners. Details about the sampling design, methodology, and questionnaires of the SHARE survey have been presented elsewhere [19]. Methods specific to this study are briefly summarized here. A population-based sample of the non-institutionalized population is used for this survey, with data collected via multi-stage sampling. We used data from wave 4 (2011–12), wave 5 (2013), wave 6 (2015) and wave 7 (2017) for both Belgium and France. Data from wave 3 (2008) were not included, as these data assessed the life histories of participants and did not provide the necessary information regarding health and behavioral risks. Moreover, due to the wide time interval between waves 1 (2004) and 2 (2006) on the one hand and wave 4 (2011) and subsequent waves on the other hand (five years), thereby resulting in interruption of the longitudinal nature of the data, we relied on data derived from waves 4 to 7.

Participants fulfilling the following criteria were included to the analysis: i) aged 50 years or above and ii) having information on the main outcome variables, heart attack, stroke or atherosclerosis at least in one wave. Data were not analyzed for participants i) < 50 years of age and ii) people who were reported as having CVD during enrollment, and iii) with no CVD information at enrollment. Overall, 7,914 adults in Belgium and 6,408 in France participated in at least one wave of the survey and were eligible, resulting in a total number of person-observations of 20,555 and 16,305 for Belgium and France, respectively.

## Outcome and exposure assessment

**Cardiovascular diseases.** The presence or absence of cardiovascular diseases was measured in the SHARE study using the question '*Has a doctor told you that you had/Do you currently have any of the conditions on this card*?'. The possible answers were: yes or no; if yes, the time of first diagnosis was asked. In this study, we defined a person as having CVD if he/she has at least one of the following problems, or both; i) a heart attack including myocardial infarction or coronary thrombosis; or ii) a stroke or cerebrovascular disease diagnosed between waves.

**Socioeconomic determinants.** Socioeconomic variables, including age, sex, household income, retirement status, marital status, educational status, and social support were collected in the SHARE study. In our study, we categorized marital status as a dichotomous variable as living with a partner or living alone, where the latter included those who were widowed, divorced or never married. Level of education was assessed with the International Standard Classification of Education (ISCED) 1997 [20], and was categorized as low (level 0–2), medium (levels 3 and 4), and high (levels 5 and 6). We categorized wealth using three quantiles for each country into upper, middle and lower one-third. We categorized living areas in three groups: i) large town or city, ii) small town, and iii) rural or village. Social support was measured using two questions: (1) '*In the past twelve months, has any family member from outside the household, friend, or neighbor given you any type of support*?' and, if 'yes', (2) '*Is there someone living in this household who has helped you regularly during the past twelve months*?'. We dichotomized this information by defining social support as having positive responses on both of the above questions, and no social support otherwise.

**Behavioral determinants.** Smoking status in the SHARE survey was assessed by asking '*Do you smoke at the present time*?'. Regular alcohol consumption was assessed with the question '*In the last three months, how often did you have six or more units of alcoholic beverages on one occasion*?'. The possible response options were: i) daily or almost daily, ii) 5 or 6 days a week, iii) 3 or 4 days a week, iv) once or twice a week, v) once or twice a month, vi) less than once a month and vii) not at all in the last 3 months. We dichotomized this variable into regular alcohol consumption (i-iv) and no regular alcohol consumption (v-vii). The frequency of

engaging in vigorous and moderate physical activity (PA) was assessed asking: '*How often do you engage in vigorous/moderate PA, such as sports, heavy housework, or a job that involves physical labor*?' The possible responses were: i) more than once a week, ii) once a week, iii) once to three times a month, and iv) hardly ever or never. We dichotomized this variable into regular PA (i and ii), and no regular PA (iii and iv). Fruit and vegetable intake was assessed by asking: '*In a regular week, how often do you have a serving of fruit and vegetables*?'. The possible responses were: i) every day, ii) three to six times a week, iii) twice a week, iv) once a week, v) less than once a week. We dichotomized into adequate (i) and inadequate intake (ii-v).

**Physical measurements and clinical characteristics.** We computed body mass index (BMI) using the self-reported values of body weight and height in each wave. Hand-grip strength was measured by trained interviewers using a handheld dynamometer. Self-reported long-term illnesses were assessed with the question '*Do you have any long-term health problems, illness, disability or infirmity such as chronic respiratory, kidney disease, cancer, etc.*?'. We categorized comorbidities into; i) no comorbidity, ii) one comorbidity, and iii) two and more comorbidities. Depression status was assessed using a EURO-12, a 12-item geriatric depression scale and those who scored below 4 were considered as not having depression and those above 4 as having depression. Moreover, whether the participant ever had CVD risks, including high blood pressure or hypertension; high blood cholesterol; and diabetes and/or high blood sugar were assessed.

## Statistical analysis

Descriptive statistics including absolute and relative frequencies for categorical variables, mean with Standard Deviation (SD) or median with Inter Quartile Range (IQR) for continuous variables were employed to summarize the characteristics of participants. We compared socioeconomic characteristics between individuals included in the analysis (n = 14,322) and those excluded as a result of missing CVD outcome values at enrollment (n = 117), using two sample t-tests and chi-square tests for numerical and categorical variables, respectively. The results show that individuals who were included are comparable with the excluded ones in terms of socioeconomic characteristics (Table 1 in S1 Annex). To estimate the population averaged effect of socioeconomic, behavioral and physical determinants on the occurrence of CVD, we developed two independent marginal models using multivariable Generalized Estimating Equations (GEEs) with the binomial distribution and logit link function [21]. GEE accounts for within-subject association and allows to estimate the between-subject effects of the relationships of determinants with the occurrence of CVD over the observation period. The unequal selection probabilities and unequal numbers of follow-up interviews among study participants was taken into consideration in the GEE analysis using individual survey weights. We assumed that the within-subject association among the vectors of repeated outcomes would have an exchangeable working correlation structure. Albeit other correlation structures were considered, goodness-of-fit indices were found under the exchangeable correlation structure (Table 3 in S1 Annex). As a goodness-of-fit indices, we used the quasi-likelihood under the Independence Model Criterion (QIC), the Wald $\chi^2$ test, and we performed a residual analysis to assess the presence of outliers and their random distribution [22, 23]. To compare the models, we used both the QIC and the Wald-test between the null, non-adjusted model and the adjusted models. Hence, the model with an exchangeable correlation structure was selected as it provided the smallest QIC. To evaluate whether the determinants vary between Belgium and France, we also performed a stratified analysis fitting a model for each country separately, however, there was no significant difference between countries in terms of covariate effects and the results are shown in Tables 7 and 8 in S1 Annex. We performed the GEE analyses using the '**geepack**' package of the free statistical software, R version 3.6.1 [24].

The percentage of missing values across all the variables varied between 0 and 37.6%. Missing results were imputed for all missing variables evaluated in the GEE model. Assuming the missing data to be missing at random (MAR), we performed Multivariate Imputation by Chained Equations (MICE) using the '**mice**' package in R [25]. We imputed the dataset to have M = 100 complete datasets using all the variables included in the model and some additional auxiliary variables (detailed description on the imputation model is available in the S1 Annex). For more details regarding the MICE approach we used, we refer the reader to a book by Van Buuren [26]. The parameters of interest were estimated in each imputed dataset separately, and combined using Rubin's rules [27]. We performed a sensitivity analysis to compare the results obtained from a multiple imputation approach with those from a complete case analysis. The results were comparable between the two approaches, except in terms of precision of the estimated model parameters. More specifically, multiple imputation was found to lead to more precise estimates which can be seen from the shorter confidence intervals (Tables 4 and 5 in S1 Annex). We assessed the distributional similarity of the observed and imputed observations using summary statistics and showed that the distributions of the imputed and observed values are comparable (Table 6 in S1 Annex).

## Ethics statement

We used publicly available data (available via the website http://www.share-project.org/data-documentation/share-data-releases.html) for a secondary data analysis. SHARE underwent a review of ethical standards by the University of Mannheim's internal review board. Ethical considerations including the written informed consent has been taken care of by another institution. Details on the conduct of the study including the ethical approval can be found elsewhere [28].

## Results

### Socioeconomic characteristics participants

Table 1 displays the socioeconomic characteristics of participants at their first enrollment. The mean age of participants was 64.4 (SD: 10.9) and 65.6 (SD: 10.9) in Belgium and France, respectively. Overall, 55.7% were females and 41.8% had low level of education. Majorities (67.0%) were living with their partners and one-third (33.1%) were living in large towns or cities. More than half (55.4%) were retired and nearly one-third (31.8%) had regular social support from family members or any other person. The median household annual net income was 32.6 and 28.6 thousand Euro in Belgium and France, respectively.

### Behavioral, physical measurements and other health related characteristics

The trend of behavioral, physical and other health related characteristics of participants at each wave of data collection is presented in detail in Table 2 in S1 Annex. Overall, 87.6% of participants reported adequate fruit and vegetable intake, i.e., at least one serving per day and the trend was consistent from 2011 to 2017. Above half (50.9%) hardly ever or never involved in vigorous physical activities, whereas, 65.6% reported that they performed moderate physical activities more than once a week. Similarly, the frequency of vigorous and moderate physical activity remains relatively constant across time. Overall, only 7.8% have a history of regular alcohol consumption within 6 months prior to data collection. Overall, one-fifth (19.9%) were smokers at the time of each interview and the proportion was highest in 2017 (29.2%). The rate of obesity is relatively consistent throughout the observation period, ranging 18.6% (2011) to 20.7% (2015). The overall mean of hand grip strength was 33.6 (SD: 11.8) and it was

**Table 1. Socioeconomic characteristics of adults aged 50 years or older in Belgium and France (n = 14,322) during enrollment in the Survey of Health, Ageing and Retirement in Europe, 2011 to 2017.**

| Participant characteristics | Total | Belgium | France |
|---|---|---|---|
| Age *(years)* | 64.9 (10.9) | 64.4 (10.9) | 65.6 (10.9) |
| Sex *(female)* | 7975 (55.7) | 4340 (54.8) | 3635 (56.7) |
| Level of education [a] (n = 14,130) | | | |
| Low | 5902 (41.8) | 3135 (40.0) | 2767 (44.0) |
| Medium | 4327 (30.6) | 2121 (27.0) | 2206 (35.0) |
| High | 3901 (27.6) | 2579 (32.9) | 1322 (21.0) |
| Marital status (n = 14,250) | | | |
| With partner | 9547 (67.0) | 5338 (67.6) | 4209 (66.2) |
| Alone | 4703 (33.0) | 2553 (32.4) | 2150 (33.8) |
| Living area (n = 13,793) | | | |
| Rural | 4965 (36.0) | 2088 (27.5) | 2877 (46.5) |
| Small town | 4263 (30.9) | 2632 (34.6) | 1631 (26.3) |
| Large town or city | 4565 (33.1) | 1631 (26.3) | 1683 (27.2) |
| Family size (no of persons) | | | |
| One | 3869 (27.0) | 2071 (26.2) | 1798 (28.1) |
| Two | 7989 (55.8) | 4326 (54.7) | 3663 (57.2) |
| Three | 1493 (10.4) | 923 (11.7) | 570 (8.9) |
| Four and above | 971 (6.8) | 594 (7.5) | 377 (5.9) |
| Estimated household income (€) (median(IQR)) (n = 14,109) | 30463.2 (31030.7) | 32640.3 (35174.8) | 28621.0 (26953.5) |
| Lower | 4404 (31.2) | 2377 (30.6) | 2027 (32.0) |
| Middle | 4452 (31.6) | 2421 (31.1) | 2031 (32.1) |
| Upper | 5253 (37.2) | 2984 (38.3) | 2269 (35.9) |
| Social support (n = 11,275) | | | |
| Yes | 3589 (31.8) | 2166 (34.7) | 1423 (28.3) |
| No | 7686 (68.2) | 4083 (65.3) | 3603 (71.7) |
| Retirement (n = 13,994) | | | |
| Yes | 7939 (56.7) | 3971 (51.7) | 3968 (62.9) |
| No | 6055 (43.3) | 3717 (48.3) | 2338 (37.1) |

a-Based on ISCED 1997 (level 0–2).

consistent across the waves. The rate of depression ranged from 29.9% in 2017 to 31.5% in 2015, with an overall rate of 30.9%.

## The prevalence of cardiovascular risks

Table 2 summarizes the prevalence of CVD risks in Belgium and France. One-third (33.2%) and 31.0% of older adults in Belgium and France, respectively, had a history of hypertension at

**Table 2. Prevalence of cardiovascular disease risks among adults aged 50 years or older in Belgium and France (n = 14,276), from the Survey of Health, Ageing and Retirement in Europe, 2011 to 2017.**

| Cardiovascular risks | Total n(%) | Belgium n(%) | France n(%) |
|---|---|---|---|
| Ever had hypertension *(Yes)* | 4596 (32.2) | 2617 (33.2) | 1979 (31.0) |
| Ever had high blood cholesterol *(Yes)* | 3898 (27.3) | 2410 (30.6) | 1488 (23.3) |
| Ever had high blood sugar *(Yes)* | 1546 (10.8) | 848 (10.8) | 698 (10.9) |
| Total | 14276 | 7888 | 6388 |

least once in their life time. The prevalence of high blood cholesterol in Belgium (30.6%) is higher than in France (23.3%). Whereas, the prevalence of high blood sugar in Belgium (10.8%) is nearly equal to the prevalence in France (10.9%).

## The occurrence of cardiovascular events

The overall rate of heart attack was 8.3%, which is 7.6% in Belgium and 9.1% in France. The rate of stroke was 2.6% and 2.3% in Belgium and France, respectively. Throughout the observation period, the overall rate of CVD was 9.5% and 10.7% in Belgium and France, respectively. (Table 3)

## Socioeconomic determinants of cardiovascular diseases among older adults

In the multivariable GEE model, the odds of having CVD was 5.7% higher for a one year increase in age [AOR: 1.057, 95%CI: 1.055–1.060]. The risk of CVD is significantly higher among those who were living in large cities [AOR: 1.14, 95%CI: 1.07–1.18] than those living in rural areas. Similarly, retired older adults showed a higher odds of CVD than those who were not retired [AOR: 1.21, 95%CI: 1.16–1.31]. Being female [AOR: 0.54, 95%CI: 0.51–0.56], high level of education [AOR: 0.82, 95%CI: 0.79–0.90], higher income [AOR: 0.82, 95%CI: 0.76–0.86] and social support [AOR: 0.81, 95%CI: 0.77–0.84] imply a significantly lower risk of CVD as compared to their respective reference categories (Table 4).

## Behavioral and physical determinants of cardiovascular diseases among older adults

In the multivariable GEE model (Table 5), on average the odds of CVD is 31% lower for those who do regular physical activity than those who do not [AOR: 0.69, 95%CI: 0.64–0.73]. Similarly, adequate fruit and vegetable intake is also associated with lower odds of CVD among older adults [AOR: 0.93, 95%CI: 0.87–0.99]. The odds of having CVD is 1.5 times higher for those who are obese than for those not obese [AOR: 1.49, 95%CI: 1.44–1.55]. Those who have one comorbidity and two or more multimorbidity are 2.2 and 4.5 times respectively, higher likelihood of having CVD than those with no comorbidity. The odds of having CVD is also 1.3 times higher for those who have depression than their counterparts [AOR: 1.27, 95%CI: 1.21–1.33]. Moreover, hand grip strength is significantly associated with lower CVD [AOR: 0.987, 95%CI: 0.984–0.990].

## Discussion

Based on a large sample of adults aged 50 years and above in Belgium and France, this study aimed to identify the socioeconomic, behavioral and physical determinants of cardiovascular diseases and the interplay in between longitudinally. Whilst several studies evidenced the role

**Table 3. Occurrence of cardiovascular diseases among adults aged 50 years or older in Belgium and France (n = 36,781 person observation points), from the Survey of Health, Ageing and Retirement in Europe, 2011 to 2017.**

| CVD events | Overall percentage | Belgium N (%) | France N (%) |
|---|---|---|---|
| Heart attack | 8.3 | 1563 (7.6) | 1483 (9.1) |
| Stroke | 2.4 | 526 (2.6) | 369 (2.3) |
| Heart attack or stroke | 10.0 | 1,954 (9.5) | 1,738 (10.7) |

CVD: cardiovascular diseases.

**Table 4. Multivariable GEE models estimating the effect of socioeconomic characteristics on cardiovascular diseases among adults aged 50 years or older in Belgium and France.**

| Variables | Percent with outcome | COR [95%CI] | AOR [95%CI] |
|---|---|---|---|
| Age | *Cont.* | 1.063 [1.059–1.067]*** | 1.057 [1.055–1.060]*** |
| Sex | | | |
| Male | 12.6 | 1 | 1 |
| Female | 8.0 | 0.60 [0.55–0.65]*** | 0.54 [0.51–0.56]*** |
| Living area | | | |
| Rural | 9.6 | 1 | 1 |
| Small | 10.4 | 1.08 [0.99–1.18] | 1.08 [0.99–1.14] |
| Large city/town | 10.5 | 1.14 [1.01–1.22]* | 1.14 [1.07–1.18]* |
| Level of education | | | |
| Primary | 12.7 | 1 | 1 |
| Secondary | 8.6 | 0.64 [0.58–0.72]** | 0.91 [0.86–0.97]* |
| Higher | 7.8 | 0.58 [0.52–0.65]*** | 0.82 [0.79–0.90]** |
| Marital status | | | |
| Partner | 9.2 | 1 | 1 |
| Alone | 11.5 | 1.30 [1.19–1.41]* | 1.04 [0.99–1.10] |
| Net income | | | |
| Lower | 12.1 | 1 | 1 |
| Middle | 10.3 | 0.86 [0.79–0.94]* | 0.99 [0.94–1.03] |
| Upper | 7.0 | 0.60 [0.55–0.67]*** | 0.82 [0.76–0.86]** |
| Social support | | | |
| No | 12.2 | 1 | 1 |
| Yes | 7.0 | 0.63 [0.58–0.69]*** | 0.81 [0.77–0.84]*** |
| Retirement | | | |
| No | 5.8 | 1 | 1 |
| Yes | 13.0 | 2.10 [1.92–2.28]*** | 1.21 [1.16–1.31]** |

* p–value<0.05

** p<0.01

*** p <0.001.

AOR: Adjusted odds ratio; COR: Crude odds ratio; GEE: Generalized Estimating Equation.

Multivariate multiple imputations were performed (n = 36,860).

• Interaction of age and level of education with living area, income and social support was assessed but statistically not significant.

of socioeconomic factors on CVD among adults, the role of these factors on older people were not well documented. This study showed that socioeconomic characteristics, including having social support, level of education, living area, retirement and income level were independent determinants of CVD among older adults.

After adjustment for the socioeconomic and other related variables, the risk of having CVD was significantly higher among older adults who live in large cities than those who live in villages. A lower green area in urban settings could be a possible reason as indicated by Seo S. and colleagues who found adults living in areas with greater amounts of green space found to have a lower risk of CVD [29]. Consistently, various studies showed that exposure to rural green space is associated with a reduction in indicators of CVD risk factors compared to urban streets [30–34]. The higher risk of CVD in the most deprived groups is more pronounced in urban areas with low amounts of green space coverage [35, 36]. Furthermore, the higher

**Table 5. Multivariable GEE model estimating the effect of behavioral and physical determinants of cardiovascular diseases among older adults in Belgium and France.**

| Variables | Percent with outcome | COR [95%CI] | AOR [95%CI] |
|---|---|---|---|
| Age | *Cont.* | 1.063 [1.059–1.067]*** | 1.045 [1.042–1.048]*** |
| **Sex** *(female)* | 8.0 | 0.60 [0.55–0.65]*** | 0.42 [0.39–0.45]*** |
| **Physical activity** | | | |
| No regular PA | 12.6 | 1 | 1 |
| Regular PA | 6.1 | 0.55 [0.51–0.60]*** | 0.69 [0.64–0.73]** |
| **Fruit and vegetable intake** | | | |
| Not adequate | 10.2 | 1 | 1 |
| Adequate | 9.3 | 0.93 [0.82–1.06] | 0.93 [0.87–0.99]* |
| **Smoking** | | | |
| No | 9.1 | 1 | 1 |
| Yes | 7.9 | 1.16 [1.02–1.31]* | 1.19 [1.13–1.25]* |
| **Regular alcohol consumption** | | | |
| No | 11.4 | 1 | 1 |
| Yes | 10.8 | 0.99 [0.80–1.21] | 1.04 [0.96–1.13] |
| **BMI** | | | |
| Normal | 8.2 | 1 | 1 |
| Overweight | 10.0 | 1.20 [1.10–1.31]*** | 1.08 [1.03–1.14]* |
| Obesity | 13.9 | 1.67 [1.50–1.85]*** | 1.49 [1.44–1.55]*** |
| **Chronic comorbidities** | | | |
| No comorbidity | 2.1 | 1 | 1 |
| One comorbidity | 8.1 | 2.84 [2.52–3.21]*** | 2.23 [2.10–2.37]*** |
| > = 2 comorbidities | 21.9 | 6.98 [6.20–7.87]*** | 4.54 [4.27–4.83]*** |
| **Grip strength** | *cont.* | 0.984 [0.980–0.988]*** | 0.987 [0.984–0.990]* |
| **Depression** | | | |
| No | 8.2 | 1 | 1 |
| Yes | 13.0 | 1.51 [1.40–1.63]*** | 1.27 [1.21–1.33]** |

* p–value<0.05

** p<0.01

*** p <0.001.

AOR: Adjusted odds ratio; COR: Crude odds ratio; GEE: Generalized Estimating Equation; BMI: Body Mass Index.

Multivariate multiple imputations were performed (n = 36,860).

Interaction of physical activity level with fruit and vegetable intake and smoking with alcohol consumption was examined but statistically not significant.

prevalence of smoking, physical inactivity, and unhealthy diet in urban areas could also be the possible reasons for higher risk of CVD. Thus, interventions aimed at reduction of CVD risks need to integrate with other sectors for optimal intervention effect.

In our study, older adults with regular social support from family or anyone else have a lower risk of CVD. This finding is similar to a study by Rosengren AL. *et al*, which indicate low social support is associated with coronary heart diseases [37]. Several studies also documented the role of social support in CVD incidence and mortality [38–40]. Both objective social isolation and the subjective perception of being isolated have been shown to be associated with a higher rate of CVD [41, 42]. Orth-Gomer explained the psychological mechanisms of social support leading to CVD morbidity and mortality [43]. Therefore, it is essential to elucidate the molecular mechanisms of loneliness leading to CVDs. Moreover, community based CVD prevention strategies need to consider the role of social support to prevent the detrimental effects of isolation.

The present study showed, retired older adults have a higher risk of CVD after adjusted for age, sex and other socioeconomic characteristics. A recent systematic review indicated the impact of retirement on the rate of CVDs and risk factors varies across countries, in which studies in the European countries showed a detrimental effect of retirement on CVDs [44]. Pedron and colleagues identified male and low-educated retirees as potential high-risk groups for worsening CVD risk factors after retirement [45]. Retirement has been linked to increased leisure time activities but, it may reduce transport and work related activities. The effect of retirement also varied with the type of job a person retired from. A study by Godard and associates showed retirement caused an increase in the likelihood of being obese among men retiring from strenuous jobs [46]. On the other hand Stenholm and colleagues indicated retirement was associated with slight weight loss in men retiring from sedentary jobs [47]. Hence, policies concerning the retirement age need to focus on ensuring whether they are suited to individuals and contexts.

In this study, higher income and higher level of education are associated with lower risk of CVDs. A multi-country study showed cardiovascular events were more common among those with low levels of education [48]. Dégano and her colleagues also indicated the rate of CVD events is 50% lower for those with university education compared to primary or lower education [49] The variation in CVD incidence and mortality based on the socioeconomic status has also been documented elsewhere [12, 14, 15, 50].

Various physical factors were also found significant in our study. Grip strength is associated with CVD after adjusting for socioeconomic and behavioral factors. Previous studies showed the association of hand grip strength with heart failure and cardiovascular risks such as high blood pressure, high blood sugar and lipid levels [51, 52]. Hand grip strength in patients with type 2 diabetes is inversely associated with CVD independently from well-established cardiovascular risks [53]. A prospective study from the UK also showed hand grip strength has inverse associations with incidence of cardiovascular events [54]. This implies hand grip strength could be a valuable CVD risk assessment tool for older adults in combination with the already available indicators.

Our findings also showed behavioral factors, including physical inactivity, inadequate fruit and vegetable intake, obesity, smoking and depression are associated with higher risk of cardiovascular events, as it is evidenced from several studies [55, 56], indicating older adults are not an exception in this regard.

The findings from this study need to be interpreted in the context of the following limitations. First, we used the existing measures from the SHARE study and were not able to include more measures of determinants that might be more appropriate to estimate the effect of behavioral determinants on CVD. For instance, we could not explore the amount of serving of fruit and vegetable intake, the duration and intensity of vigorous and moderate physical activity level, and the amount of alcohol consumption. Secondly, the measurement of socioeconomic characteristics, CVDs and risks was self-reported. We did not assess it using physical or laboratory measurements. Participants in low socioeconomic status (SES) could under-report their CVD status and risks [57], which might nullify the effect size of SES on CVD. This indicates the effect could even be more than what we found. Further studies with a more objective assessment of CVDs and risks might provide a larger effect size. Nevertheless, SHARE used cards with a list of disease conditions to probe participants, which could minimize the reporting bias. Thirdly, as our aim was not to identify a single determinant, we did not perform a comprehensive mediation analysis for each determinant-mediator-outcome relationship. We recommend future studies to identify independent socioeconomic and behavioral determinants of CVDs with extensive mediation analysis.

## Conclusions

This study addressed several aspects of cardiovascular disease prevention areas among older adults. Our research demonstrated that older adults who are retired, do not have social support, live in big cities, belong to the lowest wealth quantile and have a low educational level have a higher likelihood of CVD. It also showed that behavioral risks are prevalent in older adults living in Belgium and France and associated with an increase in CVD as documented from our study and several other studies. Physical factors such as a better hand grip strength is associated with lower incidence of CVD. The findings from this study underlined the fact that socioeconomic disparities affect the occurrence of CVDs in older adults. Therefore, community based interventions aimed at improving cardiovascular risks need to give more emphasis to high-risk older adults to get optimal benefit from the interventions. Older adults who live in large cities with no social support need to gain more emphasis to halt the continued problem of CVD and associated premature mortality.

## Supporting information

**S1 Annex. Supplementary material.** The supplementary material contains detailed information on comparison of included and excluded participants, behavioral risks overtime, sensitivity analysis of multiple imputation and complete case analysis, checking for imputation model fit, comparison of models with various correlation structure, and stratified GEE analysis for Belgium and France.
(PDF)

## Author Contributions

**Conceptualization:** Hamid Yimam Hassen, Hilde Bastiaens.

**Data curation:** Hamid Yimam Hassen.

**Formal analysis:** Hamid Yimam Hassen, Steven Abrams.

**Funding acquisition:** Hilde Bastiaens.

**Investigation:** Hamid Yimam Hassen.

**Methodology:** Hamid Yimam Hassen, Hilde Bastiaens, Kathleen Van Royen, Steven Abrams.

**Project administration:** Hilde Bastiaens.

**Resources:** Hamid Yimam Hassen, Hilde Bastiaens, Kathleen Van Royen, Steven Abrams.

**Software:** Hamid Yimam Hassen, Steven Abrams.

**Supervision:** Hilde Bastiaens, Steven Abrams.

**Validation:** Hamid Yimam Hassen, Hilde Bastiaens, Kathleen Van Royen, Steven Abrams.

**Visualization:** Hamid Yimam Hassen, Hilde Bastiaens, Kathleen Van Royen, Steven Abrams.

**Writing – original draft:** Hamid Yimam Hassen.

**Writing – review & editing:** Hamid Yimam Hassen, Hilde Bastiaens, Kathleen Van Royen, Steven Abrams.

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
