## [Decision Letter · Decision Letter 0]

23 Sep 2020

PONE-D-20-21259

Socioeconomic and behavioral determinants of cardiovascular diseases among older adults in Belgium and France: a longitudinal analysis from the SHARE study

PLOS ONE

Dear Dr. Hassen,

Thank you for submitting your manuscript to PLOS ONE. After careful consideration, we feel that it has merit but does not fully meet PLOS ONE’s publication criteria as it currently stands. Therefore, we invite you to submit a revised version of the manuscript that addresses the points raised during the review process.

ACADEMIC EDITOR:

While both reviewers found the study relevant and important, Reviewer 2 has underscored several methodological issues. Please pay careful attention to these issues as addressing them could improve the message of the paper.

We look forward to receiving your revised manuscript.

Kind regards,

Luisa N. Borrell, DDS, PhD

Academic Editor

PLOS ONE

Journal Requirements:

Reviewers' comments:

Reviewer's Responses to Questions

**Comments to the Author**

1. Is the manuscript technically sound, and do the data support the conclusions?

Reviewer #1: Yes

Reviewer #2: Partly

2. Has the statistical analysis been performed appropriately and rigorously? 

Reviewer #1: Yes

Reviewer #2: Yes

3. Have the authors made all data underlying the findings in their manuscript fully available?

Reviewer #1: Yes

Reviewer #2: Yes

4. Is the manuscript presented in an intelligible fashion and written in standard English?

Reviewer #1: Yes

Reviewer #2: Yes

5. Review Comments to the Author

Reviewer #1: The revised work is very well conducted, the ideas are original and relevant. The results are very interesting and will be very beneficial for the scientific community. Additionally, I find it well written and easy to understand. I would just like to comment that in table 1 in the Sex (Female) variable, I noticed that there is a typo: you need to close the parentheses.

Reviewer #2: The authors conducted a large cohort study evaluating socioeconomic determinants and cardiovascular disease using data collected from several waves of the SHARE study. By using marginal GEE models the investigators were able to account for between- and within- subject effects of social determinants. The paper was well thought out in many ways but there are some areas for improvement. Suggested edits/comments could be found below:

1. Please provide some analysis of individuals for which the outcomes of interest were missing. The authors state in line 108-109 that only persons with outcome data were contained in the sample but it is not clear how not including these persons may have biased the results. Even some descriptive analyses among persons with missing outcome data would be helpful. This is exceptionally important for the outcome which may be missing a large proportion of individuals who are deceased and therefore were not included in the sample.

2. Related to this point, proportions of missingness in tables 3-4 were helpful, however, in general it would be helpful to know more about the multiple imputations models. How were variables selected and were there auxillary variables used to specify the missing model?

3. Please clarify if any of the "multivariate". This appears to be used interchangeably but it seems as though "multivariable" may be the correct wording. The different types of models have very different implications.

4. While the authors allude to some of the limitations of using self-reported data, it might be helpful to include some more discussion of how measurement error may have biased some of these findings (e.g., low SES underreporting certain social behaviors such as alcohol/tobacco use/abuse, how does self-reporting CV events limit this analysis?).

5. The mediation analysis was well thought out but since this contains so many similar social constructs, the authors may benefit from removing it from the main text and supplement. It can be explored in future work when developing a causal model rather than the present predictive one. Collinearity also may be an issue between some of the crudely defined socioeconomic factors.

6. A similar analysis looking at incident CVD cases would be interesting either for this study or in the future. Is there a way to exclude persons with prevalent CVD events from earlier waves and at baseline of this study to understand if one of these factors (i.e., retirement status, depression) cause CVD in this population?

7. Reporting variability within and between countries is an advantage of using hierarchical models such as the one employed in this study but there is no mention of it in the text. Only descriptive results were presented between the countries. Please include more information on how these countries differ.

8. It will be generally important to state whether this is a predictive model that identifies risk factors and not a causal model which may be limited to confounding and selection biases.

6. PLOS authors have the option to publish the peer review history of their article (what does this mean?). If published, this will include your full peer review and any attached files.

Reviewer #1: **Yes: **MIREYA MARTÍNEZ-GARCÍA

Reviewer #2: No

---

## [Author Response · Author response to Decision Letter 0]

20 Oct 2020

Response to Reviewers

PONE-D-20-21259

Socioeconomic and behavioral determinants of cardiovascular diseases among older adults in Belgium and France: a longitudinal analysis from the SHARE study

PLOS ONE

First of all, I wish to thank the editor and the reviewers, also on behalf of all authors, for their valuable and constructive comments to improve our manuscript. We revised the manuscript based on the comments and issues raised by the editor and the reviewers. We are confident that we have addressed all these comments adequately and we therefore hope that the manuscript is accepted for publication. Find hereunder a point by point reply to the comments and questions raised by the editor and the two reviewers. 

Editor

Response: Thank you for your suggestions. We have reviewed the manuscript to meet the journal requirements and we hope that the manuscript in its current form does so.

Response: Thank you for your suggestions. We used publicly available data (available via the website http://www.share-project.org/data-documentation/data-documentation-tool.html) for a secondary data analysis. In order to clarify this, we have added the following sentences in line 208 - 214 of the revised version of the manuscript:

“We used publicly available data (available via the website http://www.share-project.org/data-documentation/data-documentation-tool.html) for a secondary data analysis. SHARE underwent a review of ethical standards by the University of Mannheim's internal review board (IRB). Ethical considerations including written informed consent has been taken care of by another institution. Details on the conduct of the study including the ethical approval can be found elsewhere (Alcser, Benson et al. 2005).”

Reviewer # 1

1. The work is very well conducted, the ideas are original and relevant. The results are very interesting and will be very beneficial for the scientific community. Additionally, I find it well written and easy to understand. I would just like to comment that in table 1 in the Sex (Female) variable, I noticed that there is a typo: you need to close the parentheses.

Response: Thank you for your feedback and appreciation for our work. We corrected the typo accordingly.

Reviewer # 2

1. The authors conducted a large cohort study evaluating socioeconomic determinants and cardiovascular disease using data collected from several waves of the SHARE study. By using marginal GEE models the investigators were able to account for between- and within- subject effects of social determinants. The paper was well thought out in many ways but there are some areas for improvement. Suggested edits/comments could be found below.

Response: Thank you for your feedback. The comments were very helpful and we revised the manuscript thoroughly. We hope that we addressed all your concerns in the revised manuscript.

2. Please provide some analysis of individuals for which the outcomes of interest were missing. The authors state in line 108-109 that only persons with outcome data were contained in the sample but it is not clear how not including these persons may have biased the results. Even some descriptive analyses among persons with missing outcome data would be helpful. This is exceptionally important for the outcome which may be missing a large proportion of individuals who are deceased and therefore were not included in the sample.

Response: Thank you for your feedback. As we mentioned in line 108 to 110, we excluded individuals who are aged < 50 years, individuals that already have CVD at enrollment and those that did not have information on the CVD outcome variables at enrollment. Participants with intermediate missingness in terms of the outcome variables are included in the analysis and managed using the MI technique. The main reason to exclude individuals with no CVD outcome information at enrollment is to make sure that the individuals that are included are free from CVD events prior to study initiation. In total, 467 participants were excluded (108 being aged below 50 years, 117 with missing outcome values at enrollment and 242 individuals that already had a history of CVD at enrollment). Although we assume missingness of CVD outcomes at enrollment to be completely at random (MCAR) by not including those 117 participants, the percentage of missing outcome data is below 0.8% and we therefore believe that the impact on the inference is limited when the underlying missingness mechanism is mis-specified. Nevertheless, we do agree with the reviewer that a descriptive analysis to deal with the baseline missingness could be explored. As recommended by the reviewer, we performed a descriptive analysis to compare the included individuals and the individuals that were excluded due to missing outcome values (n=117) in terms of a set of covariates. The descriptive results thereof are presented in table 1 of the supplementary material. The results show that individuals who were included are comparable with the excluded ones in terms of socioeconomic characteristics. We also added a description in line 166 to 171 of the revised manuscript. 

3. Related to this point, proportions of missingness in tables 3-4 were helpful, however, in general it would be helpful to know more about the multiple imputations models. How were variables selected and were there auxiliary variables used to specify the missing model? 

Response: Thank you for this comment. Indeed, we used auxiliary variables in combination with the covariate information used in the substantive model(s) in order to specify the imputation models. Examples of such auxiliary variables include family size, activities of daily living, limitation of activities, self-perceived health, among other variables. The perspective taken in the Multiple Imputation by Chained Equations (MICE) approach, also referred to as full conditional specification, is one of a specification of several imputation models which sequentially imputed missing values for a given variable when regressed against all other covariates available. For more details regarding this approach and the justification of full conditional specification, we refer the reader to (Van Buuren 2018). We revised the description in line 194 to 200 of the analysis section in order to clarify this. Moreover, more details concerning the imputation models and the covariates included therein are included in page 6 of the supplementary material. In general, the inclusion of auxiliary variables in the imputation models, next to the covariates present in the substantive model(s), albeit potentially unimportant from an explanatory perspective, leads to a more precision in terms of estimation of the model parameters in the final substantive model. 

4. Please clarify if any of the "multivariate". This appears to be used interchangeably but it seems as though "multivariable" may be the correct wording. The different types of models have very different implications.

Response: We agree with the reviewer that a multivariate analysis is different from a multiple or multivariable one, the latter being the analysis that has been conducted in our manuscript. Therefore, we revised the manuscript to consistently use ‘multivariable’ throughout the manuscript.

5. While the authors allude to some of the limitations of using self-reported data, it might be helpful to include some more discussion of how measurement error may have biased some of these findings (e.g., low SES underreporting certain social behaviors such as alcohol/tobacco use/abuse, how does self-reporting CV events limit this analysis?).

Response: We kindly agree with the reviewer that those in low socioeconomic status usually have a higher likelihood of under-reporting CVD status, risks and other health related issues, which might nullify the effect size of SES on CVD. This indicates the effect of SES on CVD could be even more than what we found. We elaborated more extensively on the limitation of the self-reported data and discussed this in view of the reviewer’s comment in line 367 to 374 of the revised manuscript.

6. The mediation analysis was well thought out but since this contains so many similar social constructs, the authors may benefit from removing it from the main text and supplement. It can be explored in future work when developing a causal model rather than the present predictive one. Collinearity also may be an issue between some of the crudely defined socioeconomic factors.

Response: We agree that the mediation analysis needs more exploration. We accept that this study and more importantly its design did not fully comply with nor allow for a formal and in-depth mediation analysis as we have a number of related determinants and mediators. Separate mediation analyses are needed, principally for each determinant-mediator-outcome relationship. We will consider to perform an extensive mediation analysis in the continuing analysis. Initially, we believed that this mediation analysis would provide some indications for future studies. However, to avoid confusion and mis-understanding for readers, we removed it from the main text and the supplement. 

7. A similar analysis looking at incident CVD cases would be interesting either for this study or in the future. Is there a way to exclude persons with prevalent CVD events from earlier waves and at baseline of this study to understand if one of these factors (i.e., retirement status, depression) cause CVD in this population?

Response: Thank you for your feedback. In fact, we excluded those individuals with CVD during the first wave of their enrollment as we mentioned in line 108 to 110 of the methods section. However, in this study, as the survey is every two years, we could not ascertain the exact number of CVD events (MI, stroke, etc..) and the exact date of such an event in between the waves. This made the incidence analysis somewhat complicated. Nevertheless, we do believe that this is very interesting and a potential avenue for further research. As suggested, we will consider to perform incidence analysis in the future.

8. Reporting variability within and between countries is an advantage of using hierarchical models such as the one employed in this study but there is no mention of it in the text. Only descriptive results were presented between the countries. Please include more information on how these countries differ.

Response: Thank you for your comment. We explored the difference between two countries extensively in the revised version. Besides the descriptive statistics, we also compared the difference between countries in several ways. Firstly, we compared both countries using differences in socioeconomic as well as the outcome variable. Secondly, as we have mentioned in line 187 to 190, we developed a separate GEE model for the two countries to assess whether the determinants vary across countries. However, the results showed no significant difference in the effect sizes of determinants between countries (detailed results are indicated in table 7 and 8 of the supplementary material). Thirdly, we performed a combined analysis without the inclusion of country-specific determinants under the assumption that such differences were unimportant based on our earlier stratified analysis. Hence, we finally preferred to present the results of the (combined) GEE model, for both countries combined. 

9. It will be generally important to state whether this is a predictive model that identifies risk factors and not a causal model which may be limited to confounding and selection biases.

Response: Thank you for your feedback. We do agree that the aim of this study is to identify risk factors for the occurrence of CVD, i.e. a so-called risk factor identification study. We stated this explicitly in the discussion section and indicated that our model is not a causal one, rather one that focuses on association and not on (the establishment of) causality. We believe that this study would provide overall insight for further individual causal study.

References

Alcser, K. H., G. Benson, A. Börsch-Supan, A. Brugiavini, D. Christelis, E. Croda, M. Das, G. de Luca, J. Harkness and P. Hesselius (2005). "The survey of health, aging, and retirement in Europe—Methodology." Mannheim Mannheim Research Institute for the Economics of Aging (MEA).

Van Buuren, S. (2018). Flexible imputation of missing data, CRC press.

Hamid Y. Hassen

Hamid.hassen@uantwerpen.be

---

## [Decision Letter · Decision Letter 1]

23 Nov 2020

Socioeconomic and behavioral determinants of cardiovascular diseases among older adults in Belgium and France: a longitudinal analysis from the SHARE study

PONE-D-20-21259R1

Dear Dr. Hassen,

We’re pleased to inform you that your manuscript has been judged scientifically suitable for publication and will be formally accepted for publication once it meets all outstanding technical requirements.

Kind regards,

Luisa N. Borrell, DDS, PhD

Academic Editor

PLOS ONE

Additional Editor Comments (optional):

You have addressed the reviewer's comments. However, I have some suggestions for the tables as they should stand alone.

Reviewers' comments:

Reviewer's Responses to Questions

**Comments to the Author**

1. If the authors have adequately addressed your comments raised in a previous round of review and you feel that this manuscript is now acceptable for publication, you may indicate that here to bypass the “Comments to the Author” section, enter your conflict of interest statement in the “Confidential to Editor” section, and submit your "Accept" recommendation.

Reviewer #2: All comments have been addressed

2. Is the manuscript technically sound, and do the data support the conclusions?

Reviewer #2: Yes

3. Has the statistical analysis been performed appropriately and rigorously? 

Reviewer #2: Yes

4. Have the authors made all data underlying the findings in their manuscript fully available?

Reviewer #2: Yes

5. Is the manuscript presented in an intelligible fashion and written in standard English?

Reviewer #2: Yes

6. Review Comments to the Author

Reviewer #2: The authors adequately addressed all comments posed to them from the first round of reviews. Thank you for providing me with an opportunity to review this work.

7. PLOS authors have the option to publish the peer review history of their article (what does this mean?). If published, this will include your full peer review and any attached files.

Reviewer #2: No

---

## [Editor Report · Acceptance letter]

26 Nov 2020

PONE-D-20-21259R1 

Socioeconomic and behavioral determinants of cardiovascular diseases among older adults in Belgium and France: a longitudinal analysis from the SHARE study 

Dear Dr. Hassen:

I'm pleased to inform you that your manuscript has been deemed suitable for publication in PLOS ONE. Congratulations! Your manuscript is now with our production department. 

Kind regards, 

on behalf of

Dr. Luisa N. Borrell 

Academic Editor

PLOS ONE